# An Agent-Based Simulation Approach for Evaluating the Performance of On-Demand Bus Services

**Sohani Liyanage *** and **Hussein Dia ***

Department of Civil and Construction Engineering, Swinburne University of Technology,
Hawthorn 3122, Australia
* Correspondence: sliyanage@swin.edu.au (S.L.); hdia@swin.edu.au (H.D.);
  Tel.: +61-39-214-4336 (S.L.); +61-392-145-280 (H.D.)

**Abstract:** On-demand multi-passenger shared transport options are increasingly being promoted as an influential strategy to reduce traffic congestion and emissions and improve the convenience and travel experience for passengers. These services, often referred to as on-demand public transport, are aimed at meeting personal travel demands through the use of shared vehicles that run on flexible routes using advanced tools for dynamic scheduling. This paper presents an agent-based traffic simulation model that was developed to evaluate the performance of on-demand public transport and compare it with existing scheduled bus services using a case study of the inner city of Melbourne in Australia. The key performance measures used in the comparative evaluation included quality of service and passenger experience in terms of waiting times, the efficiency of service and operations in terms of hourly vehicle utilization, and system efficiency in terms of trip completion rates, passenger kilometers travelled and total passenger trip times. The results showed significant benefits for passengers who use on-demand bus services compared to scheduled bus services. The on-demand bus service was found to reduce average total passenger waiting times by 89% during the Morning Peak; by 78% during the Mid-Day period; by 81% during the Afternoon Peak; and by more than 95% during other periods of the day. From an operator's perspective, the on-demand services were found to achieve around 70% vehicle utilization rates during peak hours compared to a utilization rate not exceeding 16% for the scheduled bus services. Even during off-peak periods, the occupancies for on-demand services were almost twice the vehicle occupancies for scheduled bus services. In terms of system efficiency, the on-demand services achieved a trip completion rate of 85% compared to a trip completion rate of 67% for the scheduled bus services. The total passenger-kilometers travelled was similar for both scheduled and on-demand bus services, which refutes claims that on-demand bus services induce more kilometers of travel. The trip completion times were around 55% shorter for on-demand bus services compared to scheduled services, which represents a significant saving in travel time for users. Finally, the paper presents average emissions per completed trip for both types of services and shows a significant reduction in emissions for on-demand services compared to conventional bus services. These include, on average, a 48% reduction in $CO_2$ emissions per trip; 82% reduction in NO emissions per trip; and 41% reduction in PM10 emissions per trip. These findings clearly demonstrate the superior benefits of on-demand bus services compared to scheduled bus services.

**Keywords:** Flexible Mobility on Demand (FMoD); smart urban mobility; shared mobility; agent-based modelling; sustainable public transport; disruptive mobility

## 1. Introduction

Advances in technology and emerging business models are creating new opportunities to provide urban travelers with innovative options to meet their travel needs. Today, most travelers and commuters rely either on privately owned vehicles, which offer a high degree of flexibility, independence and convenience; or they rely on public transport services, especially buses, which are considered less reliable but offer users a service that has a lower out-of-pocket travel cost compared to a privately owned vehicle [1]. However, emerging digital innovations and technologies could offer new opportunities and travel solutions that blur the line between traditional modes of transport, and can offer new options that are reliable and convenient without the high cost of owning and running a private vehicle. Rather than relying on the traditional option of using a public transport mode that runs on pre-specified schedules, new approaches offer flexible and more convenient shared on-demand vehicles that operate as public transport vehicles but do not have pre-specified timetables or schedules. Instead, they work on-demand through an app that travelers use to book their travel, helping them to access quality public transport services at a lesser cost than owning a private vehicle. While such services have been trialed in a number of cities before, the majority have failed for various reasons including insufficient information that allows for a full understanding of demand patterns, route optimization and scheduling. To help develop a better understanding of these issues and the factors required for successful commercial operations of this emerging form of public transport, this paper develops a state-of-the-art traffic simulation model to evaluate the conditions that will lead to their successful deployment and how their performance would compare to existing scheduled bus services. This work also aims to gain insights on how to make them successful for operators such that they are commercially viable while at the same time gaining public acceptance as a new form of reliable and affordable public transport.

## 2. Current Situation: Challenges and Opportunities

Even though the use of small-scale private vehicles is more attractive among travelers due to convenience, comfort and reliability, increased use of cars (or car dependency) poses many challenges environmentally, socially and economically. On the environmental level, increased use of cars leads to high emissions of toxic and harmful substances, which contribute to global warming [2,3]. Moreover, this creates the need for more car productions, extensions of road infrastructure to accommodate the increased number of cars, which challenges the scarcity of raw materials and disrupts natural habitats [1]. They also lead to many social problems such as air and noise pollution, high risk for traffic accidents and disturbing the quality of life in urban areas [4]. Majority of trips in urban environments use single-occupant vehicles which creates economic problems mainly traffic congestion, increased fuel consumption and high transport costs [5]. In Melbourne, for example, urban traffic congestion accounted for approximately $3 billion in the year 2015 which is predicted to reach its double in 2020 [6].

On the other hand, and even though public transport services introduce efficiencies and reduce reliance on private vehicles, they are still considered a less attractive option especially in outer suburbs where the quality of public transport services lag behind services in inner urban areas [7]. Issues related to lack of reliability and significant time deviations from time schedules, over-crowded carriages especially during peak hours, long waiting times at stops and public transport hubs all contribute to making them less attractive.

These issues are summarized in Figure 1. From a service operator's perspective, these conventional services, which operate on fixed routes and fixed schedules, can be cost-effective during peak periods where the fleet capacity is fully utilized. However, they are usually found to perform poorly outside peak hours, especially in low-density areas [8].

Therefore, much effort and development have gone in recent years into exploring alternatives to these two major modes of transport (i.e., private vehicles and public transport). This has been facilitated by advances and breakthroughs in technology that today allow travelers to use their mobile devices to access a range of different services such as ride-sharing and car-sharing [9–11]. The key

disadvantage with the above options, however, is that the cost of travel is still significantly higher than existing public transport options making them less attractive for urban populations. Hence, the need for on-demand shared public transport, which is seen as a middle alternative between private vehicles and mass public transport services. This mid-way approach aims to complement mass transit, and would also allow cities to develop new travel options to accommodate rising demands and also the expectations of increasingly more sophisticated transport users who demand better services and solutions in a digital age characterized by immediate responses to demand. The recent advances and technological breakthroughs have led to this novel user-centric mobile app-based approach for flexible mobility-on-demand (FMoD) services aimed as an attractive travel option and a sustainable and economically feasible alternative to conventional public transport options [12–14]. On-demand transit (also known as micro-transit) are characterized by dynamic routing, available when users request them, shared with multiple passengers, while also utilizing asset-light shuttle services [7]. Therefore, on-demand services focus on providing high quality personalized, near door-to-door service for the users, contributing to high service efficiency and performance in urban areas. This work aims to evaluate how they would work in different urban settings and identify success factors for the deployment.

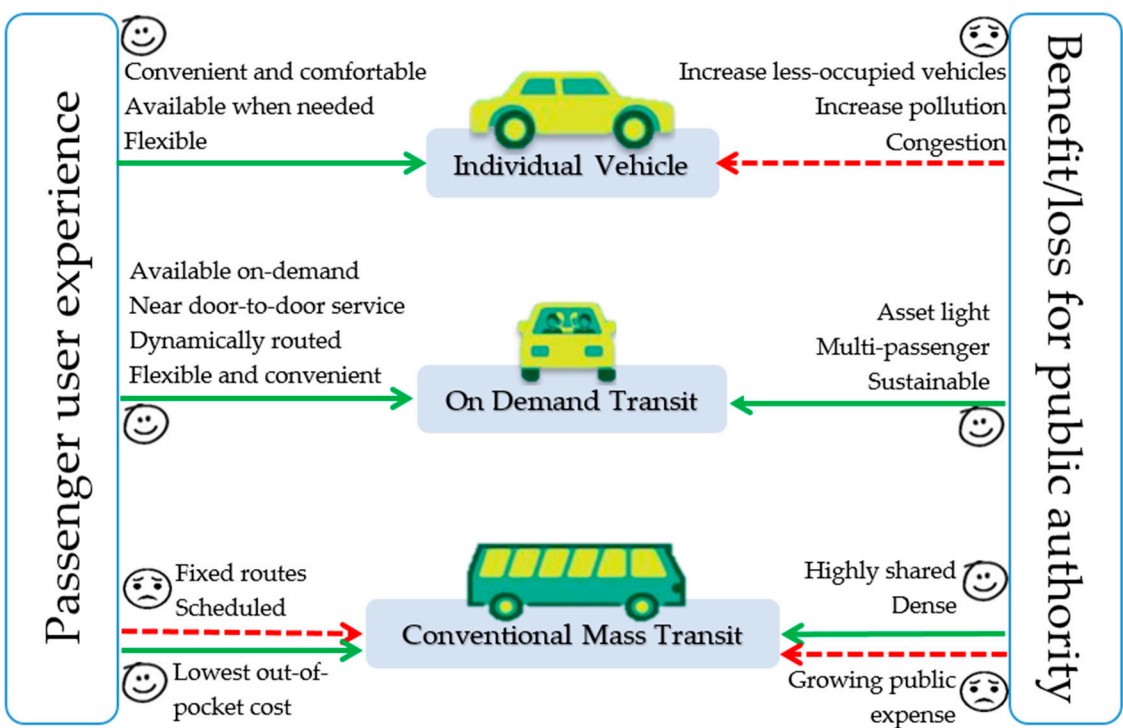

**Figure 1.** On-demand transit; a mid-way approach for customer and public needs (adapted from [7]).

## 3. Related Work

In the past, the concept of providing door-to-door services has been trialed across different cities around the world with variable degrees of success. Known as demand-responsive transport (DRT), Dial-a-Ride Transit (DART) and flexible transport services (FTS), they have evolved over the last decade or so, from operating as a service for elderly and disabled users who have difficulties in using conventional public transport to today's app-based services which allow users to book and track their trips online on a single mobile platform [15–17]. Before the widespread use of technologies and digital platforms, on-demand service schemes typically operated based on prior phone bookings (e.g., few days in advance) and manually scheduling the trip [16] in local areas where there were no or few conventional bus services. These were known as static Dial-a-Ride Transit (DART) services [18]. In today's world, emerging advanced technologies such as mobile computing, cloud and fog computing,

Internet of Things (IoT) and artificial intelligence have led to deploying new shared public transport options to improve the efficiency of urban mobility [19]. However, fully dynamic services rarely operate in practice due to the practical challenges such as finding the trade-off between the profitability of services while maintaining a quality, efficient and high-performance service to users, which are challenging objectives to achieve [15,19].

Today, there are a number of on-demand mobility service trials around the world to improve the performance and efficiency of regular bus services [19]. On-demand mobility typically takes the forms of car-sharing (short-term auto use), ridesharing (carpooling/vanpooling) and ride-sourcing services. These include on-demand single user ride-share services such as UBER, LYFT and DiDi; and car-pooling ride-share services such as Uberpool and Lyftline. Users of ridesharing service benefit from reduced trip cost compared to private vehicle travel, and in some cities, have the added advantage of being able to use high occupancy vehicle lanes which can also reduce travel times [20]. Car sharing represents short-term access to fleets of vehicles where customers are part of a community of users who pay an annual subscription fee [9,21], and are then charged based on the amount of time they have used the vehicle. Each user benefits from a personalized car service and use of a private vehicle without the costs and responsibilities of owning it [22,23]. These services are considered a serious contender to car ownership (particularly the second family car) [24] and have been reported to replace 7–10 private cars (in Bremen) and 4–6 cars (in Belgium) [25]. In another study, each car-sharing vehicle was found to have removed 9 to 13 vehicles off the road [26]. A similar study showed 30% of car-sharing users have sold a car or delayed purchasing a new one in San Francisco, USA [24,27]. In a recent study, researchers have identified one shared car is able to replace 15 private vehicles and capable of reducing car ownership by 30% [28,29]. The operation of these services is reported to be successful in more than 26 countries in 1100 cities around the world with thousands of users benefiting from such services [30,31]. Ride-sourcing services provide users with prearranged and on-demand transportation services. Users are able to book their trips and pay for the trip electronically through a smartphone app. The app also provides a platform to rate both driver and passenger [32]. In a study conducted for San Francisco, the waiting times for ride-sharing services were shorter than for taxis with vehicle utilization averaging 1.8 occupants per ride compared to 1.1 passengers for taxis [33].

Taxi services are the most conventional DART option. These operate with the highest personalized public transport service. However, they have a high cost and they do not provide a solution to encourage high occupancy vehicles on roads. Shared services, on the other hand, have flexible schedules and routes, operate on-demand and focus on delivering door-to-door services. The pioneers of this business model (namely Uber and Lyft) recently introduced options where people can share their rides with other passengers travelling in the same direction (car-pooling) [15,19]. The key success factor for these shared services is the proliferation of mobile-phone-based digital platforms where users can simply choose the best mode to travel to their desired destination and book their trips over a mobile application. The basic architecture of the on-demand bus system is illustrated in Figure 2.

Flexible on-demand public bus services are hence the most recent advancement aimed to provide a near door-to-door convenience to users at a fraction of the cost of hailing a taxi. The Helsinki experiment of on-demand public transport service included a minibus service called Kutsuplus. This service enabled the users to specify trip starting and ending points within a defined service area. An operating algorithm assigned vehicles in real-time based on passenger demand [34–36]. Another on-demand mini-bus service called BRIDJ was launched in Boston, Massachusetts and expanded to Washington DC and Kansas City in 2014. BRIDJ uses traffic and passenger data to establish trip origin-destination information in real-time to obtain the fastest routes [37]. However, the Kutsuplus service failed after a few years of operation because, in terms of the operation of Kutsuplus, operators intended to operate the service in a larger area, i.e., 100 square kilometers; however, service failed to achieve the intended user density [38]. In addition, services like Kutsuplus and BRIDJ failed due to other reasons such as low ridership, high cost, network-specific issues, lack of user awareness and limited hour availability amongst many other factors. Currently, BRIDJ has been acquired by an Australian-based transit

company that is trialing an on-demand minibus service in Sydney. The objective of this current trial is to explore the feasibility of a flexible, shared, well-coordinated transit service operation as a reliable mode of transport, providing the users with a first and last kilometer solution.

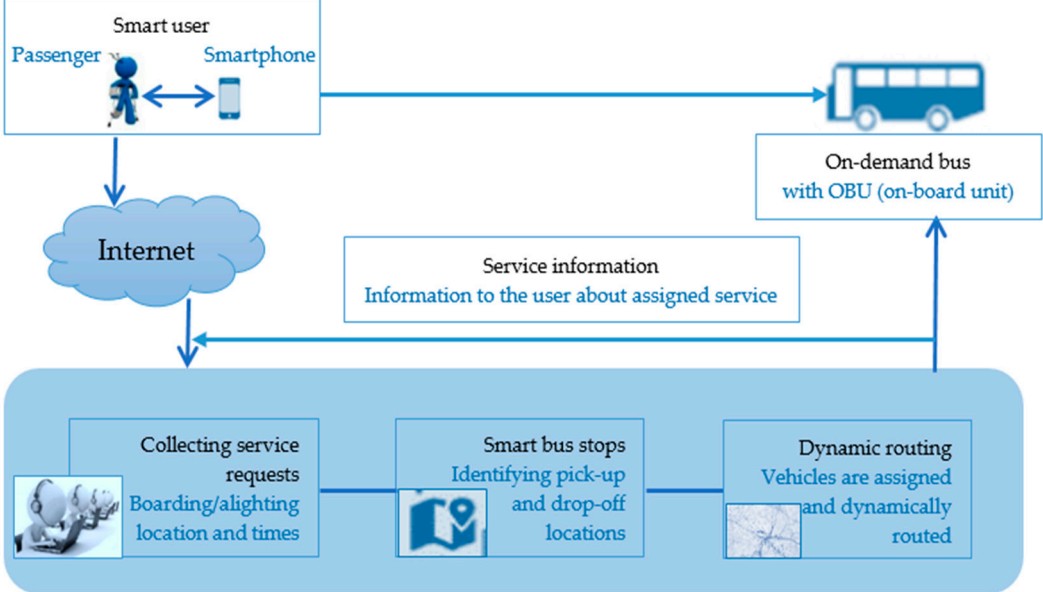

**Figure 2.** Architecture of an on-demand bus system.

To avoid failures of the past, it is important to conduct a large number of experiments and trials under variable demand conditions and network configurations. Such experiments are best undertaken and evaluate in simulation environments aimed at analyzing their feasibility and quantifying their benefits especially in comparison to existing traditional services.

This study and the results reported in this article are aimed at achieving these objectives. In particular, the contribution of this study is to understand the efficiency, performance and quality of on-demand seven-seat mini-bus service over conventional scheduled bus service.

## 4. Modelling Framework

This work adopts a microscopic traffic simulation approach to evaluate the benefits of on-demand public transport services. Simulation allows the user to generate a large number of possible scenarios and analyze whether a change to the transport network would be prohibitive or impossible. This study develops an advanced traffic simulation model, known as an agent-based simulation that can model individual travelers and their own sets of characteristics and behaviors, to explore the performance of on-demand micro-transit service and conventional mass-transit using a case study of Melbourne, Australia. This simulation model is developed using the Commuter Nano-Traffic simulator, which is an agent-based traffic simulation tool that has been utilized as the testbed for this study [39–41]. In Commuter, both travelers and vehicles are considered as agents [42–44] and travelers can be classified as a driver, a walker, a cyclist or a passenger on a train or a bus during their trips from origins to their desired destinations. Commuter generates stochastic results and uses discrete-time techniques to focus in detail of the movement of individual agents. The simulation tool is capable of keeping track of each agent and generating traveler and vehicle information over the simulation period. To travel from their origins to destinations, agents in commuter can dynamically choose the least cost route and best possible mode in real-time [39,45–47]. In the following section, some background information on microscopic traffic simulation theory is provided. Although these theories are well-established and reported in the literature [48–51], a concise section is provided next for completeness.

*4.1. Microscopic Traffic Simulation*

Microscopic traffic simulation is generally used by transport professionals to deal with dynamic and operational traffic problems and to evaluate a range of transportation and traffic engineering applications [48]. There are many applications, including on-demand transport, which are difficult to evaluate using traditional analytical tools due to the complex nature of the underlying system dynamics in these applications. With traffic simulation, the available tools provide a virtual environment where different scenarios can be introduced and evaluated in a controlled setting. Microscopic traffic simulations track the trajectory of individual vehicles in a detailed road network. Each driver/vehicle unit (DVU) is assigned a set of physical (e.g., vehicle dimension and performance) and behavioral properties (e.g., driver aggression, awareness and reaction time), which determine the motion of the vehicle and driver behavior [52]. Traffic simulations are based on different theories of microscopic traffic behavior such as car-following, lane-changing, gap acceptance and driver behavior. These models are described briefly below.

### 4.1.1. Car Following

Car following behavior has a significant impact on the accuracy of the simulation model in replicating traffic behavior on the road. Car following considers the situation of one vehicle following another in a single lane. In general, the trailing vehicle will respond to observed stimulus from the leading driver according to this relationship:

$$\text{Response} = \lambda \, \text{Stimulus} \tag{1}$$

The stimulus comprises factors such as speed, relative speed, inter-vehicle spacing, accelerations, vehicle performance and driver behavior parameters. A proportionality factor ($\lambda$) equates the stimulus function to the driver response. This relationship forms the basic philosophy behind the car following theories. Other critical parameters that govern car following models include mean headway and mean reaction time. These parameters are assigned random values for each individual vehicle according to a predefined distribution function.

### 4.1.2. Lane Changing

An understanding of the lane changing process is also necessary to allow an accurate representation of traffic flow behavior in multi-lane situations. Three main categories of lane changing models are reported in the literature: (1) The safety-based model considers the extent to which the rear vehicle in the target lane is willing to decelerate to allow the lane-changing maneuver to occur. (2) The scoring and threshold model assesses the stimulus to lane changing based on a range of scaled factors. (3) The action point model represents the human perception of speeds and distances in the lane changing process. Perceptions are compared to threshold values to determine whether drivers can achieve lane change.

### 4.1.3. Gap Acceptance and Driver Behavior

In conflicting situations, drivers must find acceptable or 'safe' gaps in which to merge with or cross the major traffic flow. Important parameters are the critical gap and follow-up headway for the minor road, and the distribution of car following headways on the major route. Some gap acceptance models include variability in the acceptable gap and have provisions for modeling drivers' tolerance thresholds as they wait for an acceptable gap. Additional driver behavior parameters, aggression and awareness, are included in some models and usually, these are assigned randomly to individual DVU according to a statistical distribution.

### 4.1.4. Traffic Modeling

Traffic simulation is characterized by a high level of modeling detail. The accuracy of the model depends on the availability and quality of the input data. After the road network is coded and the traffic demand, traffic control and public transport data are entered, the traffic conditions defined by the trip matrices are simulated. This task includes the proper selection of global modeling parameters and parameters governing car following and lane changing behavior. The modeling also includes traffic control and public transport. Traffic simulations generally take into account different types of traffic control including traffic signals and give-way signs [53]. Public transport modeling includes the definition of a public transport lines (routes, reserved lanes and bus stops), the timetables for each line (departures schedules and stop times) and the type of public transport vehicle (bus, mini-bus or tram.). This task also includes selecting the parameters of the simulation experiment. The set of parameters that characterize a simulation experiment include simulation timing, modeling, reporting and route choice. The simulation run time parameters include the specification of start and end times for the simulation in addition to the specification of the warm-up and cool-down periods. Finally, the model is run using these parameters and the simulation results are verified to determine that the model is performing as intended.

In this study, the aim is to understand the feasibility of the new on-demand services and compare their performance with conventional scheduled bus services. To undertake this, a new external plugin is activated in Commuter to add the capability of modelling buses that depart at irregular times (to differentiate them from regular bus services that operate according to schedules). The plugin helps to do the necessary calculations to add these irregularly spaced bus services into the person-route assignment model.

In Commuter, the standard logic used in conventional bus services for the expected time to wait for bus service is calculated from the interval between bus-service departures. The bus service departures are generated as the simulation progresses. For the on-demand services, every bus service departure time is calculated as soon as the model is loaded before the simulation starts. This is useful to have for situations where the public transport users can access every available public transport mode options via a single mobile platform (for this research, both scheduled and flexible on-demand bus and walking are included) and choose the lowest cost option to travel to their desired destination. In reality, their mobile app connects to a central scheduling system, which provides real-time information such as bus arrival time, bus stop location, etc. The simulation does not model these interactions between each person and the scheduling system through an app but the effects of this interactions are considered, as the simulated person will walk to the suggested bus stop and be ready for the bus when it arrives.

## 5. Study Area: Network Configuration and Travel Demands

The selected study area covers parts of four local government areas (LGAs) of Melbourne namely, Yarra, Stonnington, Port Phillip and Glen Eira districts as shown in Figure 3. The study area is around 28.9 km$^2$ consisting of 134 signalized intersections. The main reasons for selecting this study area for the case study are outlined below:

1. The study area covers major bottleneck arterials including Hoddle Street and Punt road. These corridors are identified as the busiest in Melbourne. In this first stage of evaluation, it was important to choose areas, where there are high demands as the performance of the on-demand systems, might be overstated if a low congested area had been selected.
2. The selected area's features all public transport modes available in Melbourne such as train, tram and bus. This is valuable as it allows us to model the different interactions that resemble operations across the whole city. For performance comparison purposes, this study is limited to scheduled regular size buses and on-demand mini-buses. Real operational time-schedules and routes for conventional bus services were obtained from Public Transport Victoria (PTV, 2019) [54].

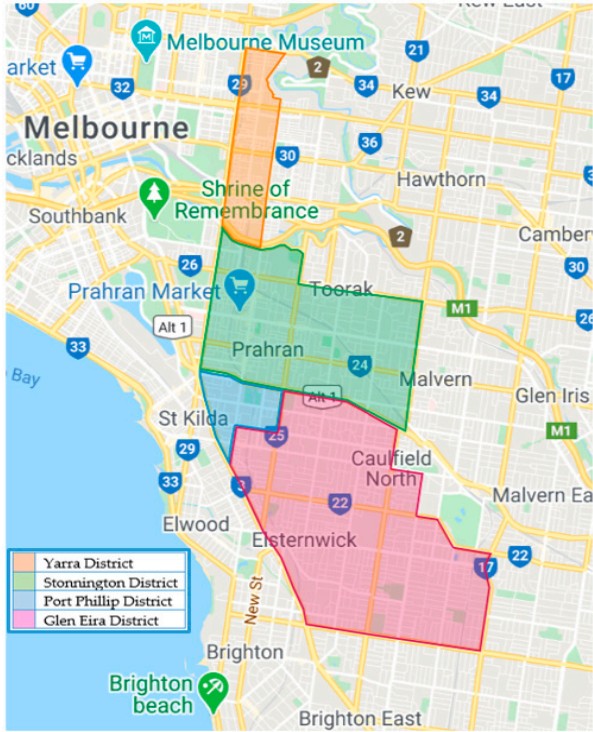

**Figure 3.** Selected study area consisting of parts of four local district areas (LGAs) in Melbourne.

As shown in Figure 3, the study area includes four local district areas (LGAs). Each LGA is subdivided into several blocks with different dimensions based on the grid-based network of Melbourne. For example, the areas of these blocks vary from 0.2 km² to 0.7 km². The length of each side of these blocks ranges from 200 m to 700 m. There is a total of 63 blocks in the selected study area, and a summary of the number of blocks for each local government area is provided in Table 1. These blocks are considered to be the trip origins and destinations of each person trip for this analysis.

**Table 1.** Number of blocks in each LGA.

| Local Government Area | Number of Blocks |
|---|---|
| Yarra District | 6 |
| Stonnington District | 18 |
| Port Phillip District | 6 |
| Glen Eira District | 33 |

## 6. Model Development

For the simulation model development, first, the transport network is coded including roads, walkways, intersections and public transport routes. The second step includes the determination and assignment of travel demands. In Commuter, traffic demands can be defined in two ways: Directed demand (origin-destination matrices) and undirected demand (origin volumes and splits). For this study, travel demands were defined using an origin-destination matrix.

Passenger demand for public transportation modes is aggregated into the block areas shown in Figure 3. For each scenario, the traffic demand is loaded to the network using a balanced origin–destination matrix. Centroids were designated to serve as the origin and destination for each person's trip. The Origin-Destination (OD) matrixes indicate the number of trips or the demand pattern among inter-block areas. For this study, OD matrices are based on the Victorian Integrated Survey of Travel and Activity (VISTA, 2019) [55], which provides an ongoing survey of travel and activity. The VISTA travel survey data used in this study is the latest available and represents the period



from May 2012 to June 2016, including 18,152 random households and 46,562 people for metropolitan Melbourne. There are 13,350 complete people trips (i.e., both the origin and destination of those trips lie within the study area) who use the bus as their usual transport mode. To develop the simulation model, more realistic background traffic is added by providing a passenger car demand of 13,912 trips. This private car demand is also based on VISTA travel survey data. The following flow chart in Figure 4 illustrates the steps involved in model development.

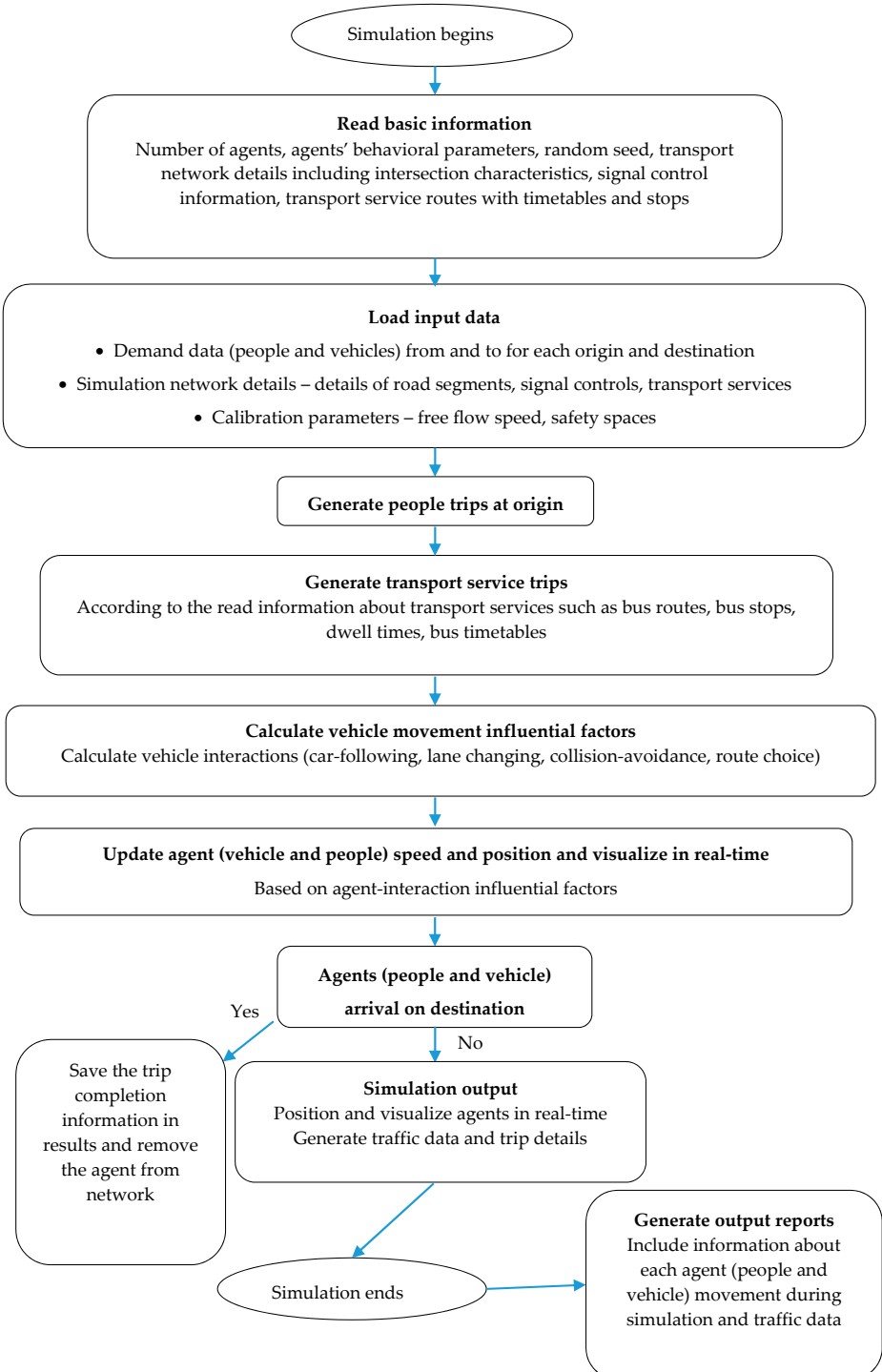

**Figure 4.** Flow chart for simulation model development.

## 7. Simulation Scenarios and Assumptions

To conduct the comparative evaluation between the existing and proposed modes of public transport, two testing scenarios are developed. The first replicates existing situation today with regular scheduled bus services using demand collected from the VISTA data. The second scenario assumed the same demand but models the on-demand mini-bus service. The qualitative separation of these scenarios is considered by varying test parameters as described below. In both scenarios, people are assumed to walk from the centroid of each origin block to the bus stand or pickup location and from the bus stand/pickup location to destination centroid of destination block. For the results reported in this article, we have assumed that the pickup locations for the on-demand services are exactly as same as existing bus stops.

It should be emphasized here that in this paper, the passenger travel demand is kept the same for both scenarios. The aim of the modelling is to uncover how the same exact demand can be met using both forms of public transport. The passenger travel demand was provided in the model using an origin–destination matrix which shows the number of passenger trips between each pair of origin and destination points. As mentioned above, this data was based on the VISTA survey data for Melbourne. For the existing bus operations scenario, the bus routes and timetables for the services are assigned according to real-world bus routes and time schedules obtained from Public Transport Victoria timetabling information.

### 7.1. Scenario 1—Scheduled Regular Bus Service

The first scenario is developed to replicate the existing situation (base case) and simulate the conventional bus services, which operate on fixed routed and fixed timetables. This scenario represents the existing conditions where people use regular static scheduled buses to reach their destinations. As mentioned in previous sections, the real-world bus routes, bus stops and bus timetables are downloaded from the PTV, 2019 [54] and used in the development of this model. Fifteen bus routes cover the study area (a total of 30 bus services when both inbound and outbound services are taken into consideration). The selected bus fleet is a regular diesel two-door bus with a capacity of 48 passengers seated plus 16 standing capacity.

### 7.2. Scenario 2—On-Demand Mini-Bus Service

The second scenario is developed to simulate how the proposed new on-demand buses would work by using the same transport network and the same base demand used in Scenario 1. The seating capacity of the on-demand minibus is assumed to be seven passengers, all seated. The development of this scenario would allow for direct comparison of the impacts of the two services and including vehicle occupancy (number of passengers), vehicle utilization (total number of passengers divided by available capacity), passenger waiting times, passenger travel times and total travel distances using each type of service. The testing parameters for this scenario are the same as the base scenario except for the fleet size and bus schedule. Bus scheduling for on-demand services was developed based on the demand profiles for the time of the day and bus patronage data for each fixed real-world route from VISTA data for the year 2014–2015 [55].

Note that the same passenger origin-destination matrix based on VISTA data is used for both scenarios with same bus stops and bus routes. The simulation period of the testing scenarios is considered from 04:00 to 00:00 on a regular weekday. The aim is to compare vehicle occupancies, passenger-waiting times and travel times, and understand how on-demand buses would compare to conventional bus services.

## 8. Simulation Results

A comparative evaluation between the two scenarios was undertaken using a number of key performance measures which included quality of service to travelers in terms of waiting and travel times, and also in terms of operational measures such as vehicle occupancy rates and utilization.

While there is a large number of parameters that can be reported and contrasted, only a number of key performance indicators that have been reported in the recent literature about the benefits of on-demand services were considered. These indicators mainly attribute the benefits of on-demand services to be in terms of reductions of passenger wait time at stops, increase in hourly vehicle utilization rates, increase in trip completion rates and the overall reduction of travel time. Hence, the focus was on these indicators to determine whether the potential benefits that are widely reported in the literature can be corroborated with the results derived from the simulation.

### 8.1. Quality of Service and Passenger Experience

The quality of service is evaluated in terms of passenger waiting times at bus stops or pick up locations, and the total travel times.

Waiting Time at Stops

Waiting times at stops include two components. The first component, T1, is the waiting time for the arrival of first bus service, which is the time from when a passenger arrives at the bus stop to the arrival time of the first vehicle of the service, averaged over all passengers simulated in the model. The second component, T2, is the time from when a passenger arrives at the bus stop to the time when the passenger is on-board the service, also averaged over all passengers simulated in the model. This differentiation is important because sometimes (especially with the scheduled bus services) the first bus that arrives is either late or full, or there are a large number of passengers waiting at the bus stop resulting in some passengers not finding room to board the bus.

The simulation period was set between 5:00 a.m. to midnight and was divided into six periods to provide more insights into the quality of the services (Table 2 and Figure 5).

**Table 2.** Passenger waiting time at stops.

| Waiting Time for a Bus (minutes) | | T1 | T2 | Total |
|---|---|---|---|---|
| Early Morning 5:00 a.m. to 7:00 a.m. | Scheduled | 16.9 | 2.01 | 18.9 |
| | On Demand | 0.3 | 0.7 | 1.0 |
| | % difference | 98% | 67% | 95% |
| Morning Peak 7:00 a.m. to 9:00 a.m. | Scheduled | 6.0 | 2.7 | 8.7 |
| | On Demand | 0.2 | 0.8 | 1.0 |
| | % difference | 97% | 71% | 89% |
| Mid-Day 9:00 a.m. to 3:00 p.m. | Scheduled | 9.6 | 2.1 | 11.8 |
| | On Demand | 0.5 | 2.1 | 2.6 |
| | % difference | 95% | −0.6% | 78% |
| Afternoon Peak 3:00 p.m. to 6:30 p.m. | Scheduled | 6.7 | 2.1 | 8.7 |
| | On Demand | 0.1 | 1.6 | 1.7 |
| | % difference | 98% | 24% | 81% |
| Evening 6:30 p.m. to Mid-night | Scheduled | 138.8 | 40.8 | 179.6 |
| | On Demand | 0.9 | 6.8 | 7.7 |
| | % difference | 99% | 83% | 96% |

T1:　Time from when a passenger arrives at the bus stop to the arrival time of the first vehicle of the service, averaged over all passengers simulated in the model.

T2:　Time from when a passenger arrives at the bus stop to the time when the passenger is on-board the service, also averaged over all passengers simulated in the model.

The values in Table 2 and Figure 5 are based on average values from a large number of different simulation runs in which the traffic is generated based on random seed numbers. When trips are generated in the model, a random seed with demand weight of 100% is selected. The departure times are designated as "exact-stochastic" which means that for each trip from an origin A has a departure time chosen at random. For example, if 600 trips are specified to depart from zone A and arriving at zone B in one hour, then exactly 600 trips will be generated between A and B, but their departure times will be irregular. Therefore, these results are not single values generated from one set of values. Rather, they are averages under varying conditions and hence can be treated with good confidence.

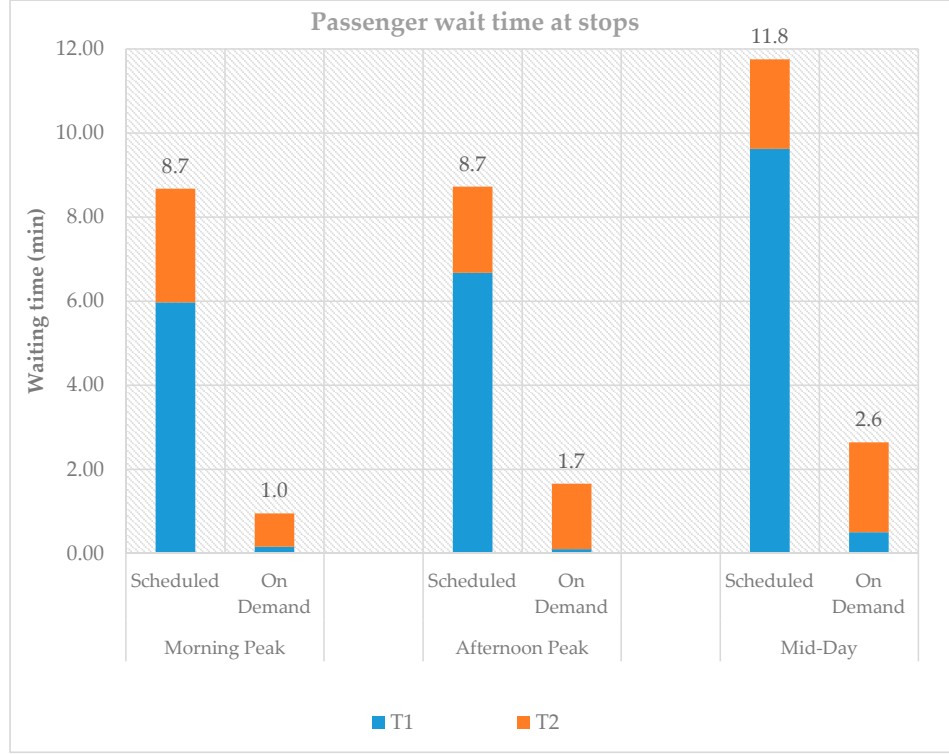

T1: Time from when a passenger arrives at the bus stop to the arrival time of the first vehicle of the service, averaged over all passengers simulated in the model.
T2: Time from when a passenger arrives at the bus stop to the time when the passenger is on-board the service, also averaged over all passengers simulated in the model.

**Figure 5.** Average passenger waiting time at bus stops.

The six time-periods included Early Morning (5:00 a.m. to 7:00 a.m.), Morning Peak (7:00 a.m. to 9:00 a.m.), Mid-Day (9:00 a.m. to 3:00 p.m.), Afternoon Peak (3:00 p.m. to 6:30 p.m.) and Evening (6:30 p.m. to midnight). These intervals were based on observations of peak and non-peak travel habits in the study area. Inspection of the results in Table 2 and Figure 5 provide a number of insights which are summarized next:

- Early Morning period shows a significant reduction (95%) in the average total passenger waiting times from 18.9 minutes for scheduled buses to 1 minute for on-demand buses;
- In the Morning Peak period, passengers who use scheduled buses experienced an average total waiting-time of 8.7 minutes, compared to an average total waiting time of 1 minute if they had used the on-demand service. This is equivalent to 89% reduction in the total waiting time, which is a significant advantage for the on-demand services during peak periods;
- The Mid-Day results also show that using on-demand bus services provide a reduction of 78% in the average total passenger waiting times. The average total passenger waiting time was 11.8 minutes for scheduled buses, compared to 2.6 minutes for on-demand mini-bus option;

- The Evening Period results show that using on-demand bus services provided a reduction of 96% in the average total passenger waiting times. The average total passenger waiting time was 179.6 minutes for scheduled buses, compared to 7.7 minutes for on-demand mini-bus option. The high waiting times for scheduled buses are the result of fewer scheduled bus services available during the Evening Period. During that period, some routes have no scheduled bus services and on other main routes, the frequency of services is reduced.
- The Afternoon Peak period showed a significant reduction (81%) in the average total passenger waiting times from 8.7 minutes for scheduled buses to 1.7 minutes for on-demand buses;

In summary, these results show significant benefits for passengers who use on-demand bus services compared to scheduled bus services. The on-demand bus service was found to reduce average total passenger waiting times by 95% during Early Morning period; by 89% during the Morning Peak; by 78% during the Mid-Day period; by 81% during Afternoon Peak; and by 96% during the Evening period.

### 8.2. The Efficiency of Service (Operator Perspective)

From a bus operator's perspective, it is important that their vehicle fleets are well utilized to improve the cost-effectiveness of service operations. One of the relevant measures used in this study is vehicle occupancy which is the average number of passengers that each vehicle carries per hour.

Hourly Vehicle Utilisation (Passengers per Hour)

The hourly vehicle utilization rates for the scheduled and on-demand bus services are presented in Figure 6. These are calculated by dividing the number of passengers on-board (occupancy) by the total vehicle capacity, averaged every hour. For the scheduled bus services, the vehicle occupancy during peak ranged between 10% and 16%. This is significantly lower than vehicle occupancies for on-demand mini-bus services that were found to achieve around 70% occupancy during peak hours. Even during off-peak periods, the occupancies for on-demand services were almost twice the vehicle occupancies for scheduled bus services

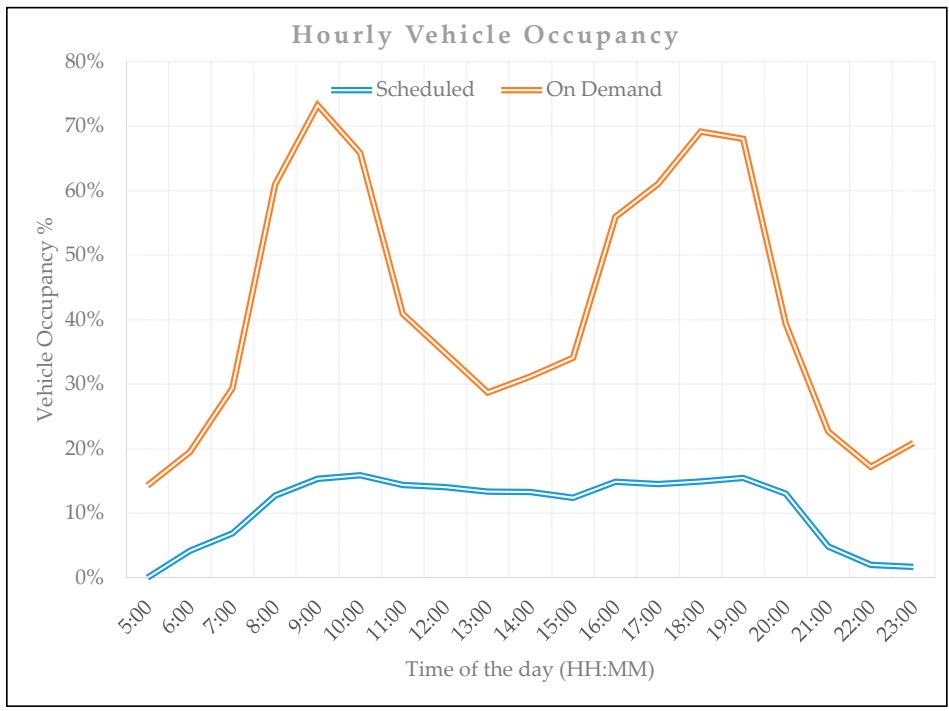

**Figure 6.** Hourly vehicle occupancy for scheduled and on-demand bus services.

*8.3. System Efficiency (Public Benefits)*

This set of performance measures refer to the system-wide impacts and efficiencies achieved through each of the modes being evaluated. These measures include how well each mode of transport performs in terms of trip completions, total passenger kilometers travelled and passenger trip times.

### 8.3.1. Trip Completion Rates

Trip completion rate is the percentage of completed passenger trips during the simulation period. In other words, it is a measure that shows whether all travelers who wanted to use the bus services were able to reach their destination and complete their trips within the simulation period. At the start of a simulation run, a total of 13,550 passenger trips were assigned to each of the scheduled and on-demand bus scenarios. The simulation period was 5:00 a.m. to midnight, including one-hour warm-up and cool-down periods at the start and end of the simulation, respectively.

In the scheduled bus scenario, the results showed that all of the 13,550 trips departed during the simulation but only 9,025 trips arrived at their desired destinations at the end of the simulation (a trip completion rate of 67%). This is due to the fact that some route services stop early in the day (e.g., some routes do not operate after 3:00 p.m.). Some other passenger demands after 8:00 p.m. were also not fulfilled due to the lack of services. For the on-demand bus scenario, however, a total of 11,491 trips (out of the 13,550 trips simulated) arrived at their destination by the end of the simulation (a trip completion rate of 85%). While these findings show that 24-hour demands are difficult to meet using either type of service, the results clearly show that on-demand services were able to deliver 85% of trips to their destinations, compared to only 67% for the scheduled buses, which is a significant benefit in favor of the on-demand bus services.

### 8.3.2. Passenger Kilometres Travelled

Passenger kilometers travelled (PKT) is the sum of the total distance travelled by all passengers who completed their trips within the simulation period. This includes passengers' walking distance from the origin centroid to the nearest bus stop; distance travelled in the bus; and walking distance from the bus stop to destination centroid. The PKT results are presented in Figure 7 and show no significant difference between scheduled and on-demand bus scenarios because both services followed common bus routes. The slight differences in Figure 7 are attributed to different walking distances to the nearest bus stops. For example, fixed schedule buses do not run after certain times resulting in fewer travelers walking to the bus stops to use the service. For the on-demand services, however, they are available at all times meaning there would be more travelers walking to pick-up locations resulting in more walking distances. As shown in Figure 7, passengers (on average) travelled about 9 km during peak hours and 7 km during off-peak hours. The daily averages were 6.4 km for the scheduled services, compared to 7.1 km for the on-demand services.

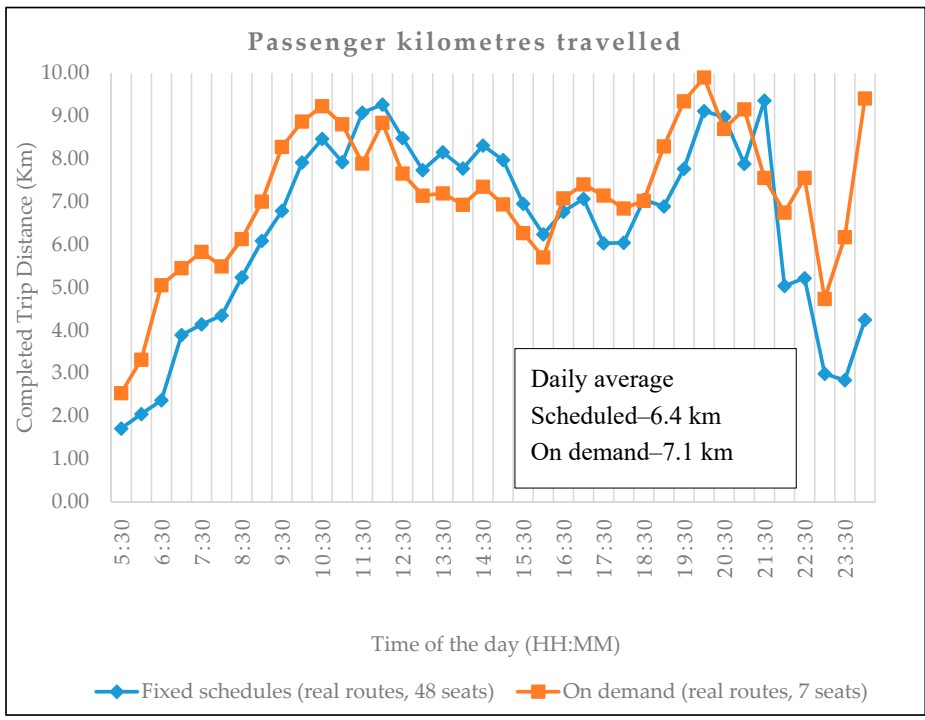

**Figure 7.** Passenger kilometers traveled.

### 8.3.3. Completed Trip Time

Completed trip time (CTT), or passenger-minutes travelled, is the sum of total travel time experienced by all travelers who completed their trips within the simulation period. This includes walking time, in-vehicle travel time and waiting time at stops and road crossings. The CTT results are presented in Figure 8 and show a significant benefit for on-demand services compared to scheduled buses. These results show that on-demand services can meet the same demand for travel while reducing the total passenger trip times. Figure 8 shows that during off-peak periods (e.g., 10 a.m. to 5 p.m.), the trip completion times for schedule buses are very high with large variability due to a long passenger waiting times as a result of running less frequent bus services. Figure 8 shows how on-demand bus services provided less variability in trip completion times making them more reliable for travelers to use. It is also interesting to point out that the on-demand bus services provided less variability when comparing peak and non-peak trip completion times allowing travelers to reach their desired destinations in the shortest amount of time compared to scheduled services. The daily average CTT was 112.5 minutes for scheduled bus services, compared to 50.1 minutes for the on-demand services (a reduction of 55% in CTT for on-demand services).

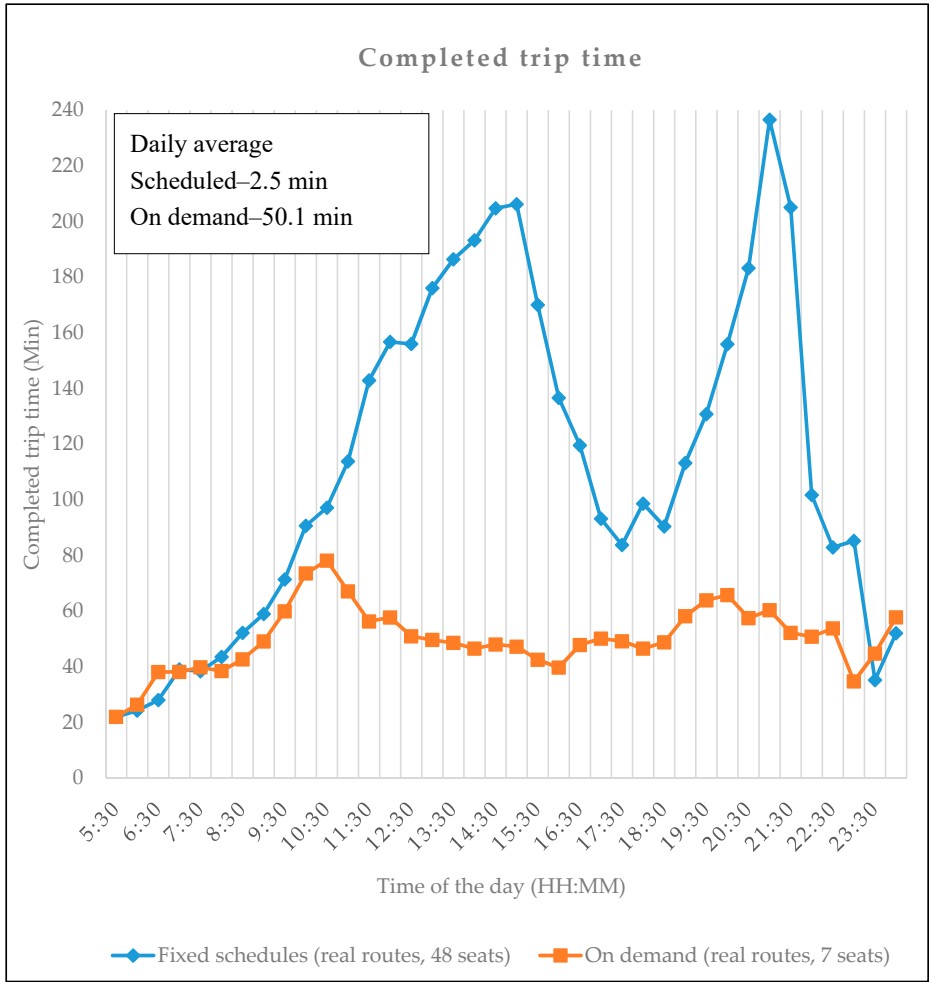

**Figure 8.** Passenger trip time.

### 8.4. Environmental Emissions

The final part of the analyses was to consider and compare the environmental emissions [56,57] for both types of services. The analysis included $CO_2$, NO and PM10 emissions for completed trips. The current conventional buses were modelled as diesel buses (reflecting a majority of the current bus fleet in Melbourne). For the on-demand services, we selected the emissions parameters for a seven-seater people mover (which is the nearest to the size of vehicles we have modelled in our scenarios). To make the comparison valid, both services were assumed to run using diesel vehicles. The results provided in Table 3 below include average emissions per completed trip and show a significant reduction in emissions for on-demand services compared to conventional bus services. These include, on average, a 48% reduction in $CO_2$ emissions per trip; 82% reduction in NO emissions per trip; and 41% reduction in PM10 emissions per trip.

**Table 3.** Average emissions per completed transport trip.

| Simulation Scenario | Conventional Scheduled Services (Diesel Buses) | On-Demand Services (Diesel 7-Seater People Mover) | Reductions in Emissions Attributed to On-Demand Services |
|---|---|---|---|
| $CO_2$ (kg) | 2.12 | 1.11 | −48% |
| NO (g) | 20.96 | 3.77 | −82% |
| PM10 (g) | 0.59 | 0.35 | −41% |

### 9. Conclusions and Future Directions

This work developed and evaluated a model for flexible on-demand mini-bus public transport services and analysed the quality, performance and efficiency of the service and compared it to the conventional fixed time scheduled bus services using a case study for the inner suburbs of Melbourne, Australia. The comparative evaluation was based on a number of performance measures which included (1) quality of service and passenger experience; (2) efficiency of service (from operator's perspective) in terms of hourly vehicle utilization rates; and (3) system efficiency and public benefits including trip completion rates, passenger kilometres travelled and completed trip times.

The comparison of the quality of services and passenger experience was analysed based on the waiting times for passenger pick-up. The results showed significant benefits for passengers who use on-demand bus services compared to scheduled bus services. The on-demand bus service was found to reduce average total passenger waiting times by 95% during Early Morning period; by 89% during the Morning Peak; by 78% during the Mid-Day period; by 81% during Afternoon Peak; and by 96% during the Evening period.

Secondly, the efficiency of both services was compared by analysing hourly vehicle occupancies and utilisation rates. For the scheduled bus services, the vehicle occupancy during peak ranged between 10% and 16%. This is significantly lower than vehicle occupancies for on-demand mini-bus services that were found to achieve around 70% occupancy during peak hours. Even during off-peak periods, the occupancies for on-demand services were almost twice the vehicle occupancies for scheduled bus services.

Thirdly, the system efficiency and public benefits were compared by analyzing trip completion rates, passenger kilometers of travel and completed trip times. For trip completion rates, the results showed that on-demand services were able to deliver 85% of trips to their destinations, compared to only 67% for the scheduled buses, which is a significant benefit in favor of the on-demand bus services. For passenger kilometers of travel, the results did not show any significant differences between scheduled and on-demand bus scenarios. While more walking was observed for the on-demand services during late evenings, these were the result of users being able to travel during periods when the scheduled bus services were not available. For the completed trip times, the results showed less variability in trip completion times for on-demand services making them more reliable for travelers to use. The on-demand bus services also provided less variability when comparing peak and non-peak trip completion times allowing travelers to reach their desired destinations in the shortest amount of time compared to scheduled services. The daily average completed trip time was 112.5 minutes for scheduled bus services, compared to 50.1 minutes for the on-demand services (a reduction of 55% in favor of on-demand services).

Finally, the paper presented the average emissions per completed trip for both types of services and showed a significant reduction in emissions for on-demand services compared to conventional bus services. These include, on average, a 48% reduction in $CO_2$ emissions per trip; 82% reduction in NO emissions per trip; and 41% reduction in PM10 emissions per trip.

In summary, this work's contributions to knowledge include identification of a new framework for simulating on-demand public transport and development of key performance measures for evaluating their impacts and benefits. This work will also have policy implications and the findings will be of direct interest to decision and policymakers and regulators. These findings will help them better understand the implications of on-demand services and the benefits and cost reductions and efficiencies that can be introduced through digital innovations.

In terms of future research directions, the authors are exploring improvements that can be achieved by optimising service capacities to meet desired hourly vehicle occupancy rates that would make the on-demand services commercially viable and sustainable while maintaining reduced waiting times for passengers. The authors are also currently undertaking stated preference surveys that will provide insights into the socio-economic and travel characteristics of users in the study area and their propensity to use the on-demand services compared to their existing travel mode choices. Our traffic

simulation models are currently being updated with new data that has been obtained from MyKi cards (smart cards used for public transport in Melbourne). The use of smart card data that show passenger touch-on and touch-off will provide a more detailed understanding of existing passenger demand profiles and patterns, which would allow for more realistic simulation and more reliable results. Once the base travel demands are established from such data, and the stated preference surveys are completed, machine learning and neural network methodologies will be used to estimate future travel demands for the on-demand bus services, which will then be used in our models to optimise fleet sizes and maximise occupancy rates.

**Author Contributions:** The authors' contributions are as follows: Conceptualization, S.L. and H.D.; Methodology, S.L.; Writing—original draft preparation, S.L.—review and editing, H.D.; supervision, H.D. All authors have read and agreed to the published version of the manuscript.

**Acknowledgments:** Sohani Liyanage acknowledges her PhD scholarship was provided by the Swinburne University of Technology.

**Conflicts of Interest:** The authors declare no conflict of interest.

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
