# Peer review of "An Agent-Based Simulation Approach for Evaluating the Performance of On-Demand Bus Services"

_sustainability, doi:10.3390/su12104117_

Round 1
Reviewer 1 Report
- The topic of this paper is interesting. However, some minor points remain to revise and improve.
- Page 3 indicates that Fig. 1 is adapted from Hazan et al. (2019). However, if it is a direct copy and page Hazan et al. (2019), it is usually not legally allowed. It is suggested that the authors should draw a different diagram by citing Hazan et al. (2019).
- Page 5 only briefly introduced the modeling framework. However, there is no single equation in the model listed here. Pages 6-13 directly report the numerical results without an explicitly listed model. It is indeed not clear how these numerical outcomes are obtained. It is strongly suggested that a least some important equations in the mathematical model should be listed and well explained.
- What is the software package sued for numerical simulations in this paper?
- What is the solution concept used for this agent-based model? Is the equilibrium of the model unique or multiple.
- Some areas in Melbourne are selected for the numerical simulation, which are interesting examples with numerical outcomes. However, it would be more informative to compare differences in the current situation and the numerical suggestions.
- Since this journal focuses on sustainability, it will be more informative if more environmental benefits made by the changing from the status quo to the numerical solutions are further addressed.
- The formats of references are seriously inconsistent. For instance, some authors are missing (e.g., [12] and [18]); some full author names are shown (e.g., [7]) while others use abbreviated given names (e.g., [17]); and some journal or book names are missing (e.g. [7]). The authors should carefully make all of the formats of references consistent.
- In addition to the application to Melbourne’s case, what kinds of academic research hypotheses are empirically tested in this paper? How about the managerial and policy implications in this paper?
Reviewer 2 Report
The topic of the research is very important for future low carbon transportation system in many urban cities. And it is good thing to show the comparison between the current existing bus system and new on-demand buss system. However the modelling and simulation technology explanation are literary expression in this paper. Therefore it is difficult to understand the confidence of the simulation results. So I strongly suggest to add and more describe about new on-demand buss system mechanism and detail simulation methodology in your paper. It is good idea to use flow chart and or simulation steps how to calculate and get the simulation results.
The following is small improvement idea.
1) Figure 1 does not need for on-demand buss system explanation. If you need to explain its system, it is better to show more on-demand buss architecture pictures.
2) In figure 2, you need to explain more with the relationship with Table 1. There is no detail explanation in Figure 2 such as several dots, travel path matrix, and some district names.
3) It is necessary to explain about confidence of the value of Table 2 simulation result.
4) In page 7 and 6, it is necessary to explain about new on-demand buss system architecture and simulation method, which I mentioned in the above comments.
Good luck.
Reviewer 3 Report
Major comments and remarks:
- Replacement rate of car-sharing services (row 134-139) should be provided in a more complex and updated form. An example from Bremen and Belgium comes from presentation/paper published in 2005. Also further example of the US.S was published in 2011. There is extensive literature on this topic and Authors present recent findings.
- Also explanation of Kutsuplus service cancellation in 2015 is not fully explained. There were many different reasons behind its cancellation. They are widely described (i.e.: http://sharedusemobilitycenter.org/news/killed-kutsuplus-3-takeaways-cities-pursing-mobility-demand/ )
- In rows 173-175 Authors claim that "In particular, the contribution of this study is to understand the efficiency, performance, and quality of on-demand 7-seat mini-bus service over conventional scheduled bus service". As we can say that performance and quality of on-demand service was presented, it is difficult to find direct link to the efficiency. Shouldn't the efficiency category be assessed using the supply of both assessed sub-schemes?
- (rows 262-286) Scenario descriptions should be more detailed. There is no information about supply of public transport and on-demand mini-bus service.
- (rows 288-290) Authors write that "A comparative evaluation between the two scenarios was undertaken using a number of key performance measures which included quality of service to travelers in terms of waiting and travel times, and also in terms of operational measures such as vehicle occupancy rates and utilization" but do not provide any explanation why those key performance measures were selected for this evaluation. In public transport theory and practice there are many other key performance indicators that might be used for this evaluation.
Minor formal remarks
6. Text in rows 227-228 lacks of (probably) reference to concrete figure
7. Similar remark is for the text between rows 240-241
8. Some references are not complete - lack of publisher (i.e ref. nr 7)
Round 2
Reviewer 1 Report
- The authors have adopted all suggestions from this reviewer and made revisions in accordance.
- However, in the revised version there are still some typos, grammar, format problems. For instance, on page 2 it shows “Error! Reference source not found.” Page 11 has a lot of empty space. On page 18, it should be “help them better understand” instead of “help them to better understand.” Professional English editing by a native English speaker may still be needed.
Reviewer 2 Report
The section 4 is very good to introduce the simulation style and method. And several changes and modification are also good approach.
However it is still difficult to evaluate how to reach the results based on this mapper because there is no detail simulation method or tool explanation. There is the flow chart in Figure 4. From Figure 4, it is still unknown how to get and or simulate the result number. This is important to show your simulation accuracy by other researchers. Therefore this lack of technical simulation and or method makes the difficulty to judge this result is reasonable or not.
This time authors added another discussion point such as the environmental emission. In terms of the emission discussion, it requires emission factor which is used to calculate the reduction value. In this topics, there are so many different emission factor for each transportation such as what kind of vehicles with how many number of passengers and how many time to stops in one route driving. Therefore I recommend if authors want to add the emission factor topics, it should be separated from this manuscript as another manuscript.
I also would like to mention it is necessary to describe how dynamic routing is achieved in this study because there are many on-demand bus services in the world, So it should be mentioned what is the different and unique point compared with other on-demand bus services. I am not able to get this point clearly from this manuscript.
Reviewer 3 Report
The paper was extensively improved by Authors. Therefore i have no more suggestions and i accept as it was provided in a modified version.
